# Interpreter usage and associations with latent tuberculosis infection treatment acceptance and completion in the USA among non-U.S.– born persons, 2012–2017

Rebeca Gonzalez-Reyes[1], Dolly Katz[2], Lauren Lambert[2], Yoseph Sorri[3], Masahiro Narita[3,4], David J. Horne [4]*, for the Tuberculosis Epidemiologic Studies Consortium[¶]

1 University of Washington, Seattle, Washington, United States of America, 2 Centers for Disease Control and Prevention, Atlanta, Georgia, United States of America, 3 TB Control Program, Public Health–Seattle & King County, Seattle, Washington, United States of America, 4 Division of Pulmonary, Critical Care, & Sleep Medicine, University of Washington, Seattle, Washington, United States of America

¶ Membership of the Tuberculosis Epidemiologic Studies Consortium is listed in the Acknowledgments.
* dhorne@uw.edu

**Data Availability Statement:** The data is now publicly available at the CDC's website: https://data. cdc.gov/National-Center-for-HIV-Viral-Hepatitis-

## Abstract

### Background

Latent tuberculosis infection (LTBI) screening and treatment interventions that are tailored to optimize acceptance among the non-U.S.–born population are essential for U.S. tuberculosis elimination. We investigated the impact of medical interpreter use on LTBI treatment acceptance and completion among non-U.S.–born persons in a multisite study.

### Methods

The Tuberculosis Epidemiologic Studies Consortium was a prospective cohort study that enrolled participants at high risk for LTBI at ten U.S. sites with 18 affiliated clinics from 2012 to 2017. Non-U.S.–born participants with at least one positive tuberculosis infection test result were included in analyses. Characteristics associated with LTBI treatment offer, acceptance, and completion were evaluated using multivariable logistic regression with random intercepts to account for clustering by enrollment site. Our primary outcomes were whether use of an interpreter was associated with LTBI treatment acceptance and completion. We also evaluated whether interpreter usage was associated treatment offer and whether interpreter type was associated with treatment offer, acceptance, or completion.

### Results

Among 8,761 non-U.S.–born participants, those who used an interpreter during the initial interview had a significantly greater odds of accepting LTBI treatment than those who did not use an interpreter. There was no association between use of an interpreter and a clinician's decision to offer treatment or treatment completion once accepted. Characteristics associated with lower odds of treatment being offered included experiencing homelessness

STD-and-TB/Tuberculosis-Epidemiologic-Studies-Consortium-TBES/5hpj-p74g/about_data.

**Funding:** This work was supported by a contract with the Centers for Disease Control and Prevention. All authors report no potential conflicts of interest.

**Competing interests:** The authors have declared that no competing interests exist.

and identifying as Pacific Islander persons. Lower treatment acceptance was observed in Black and Latino persons and lower treatment completion by participants experiencing homelessness. Successful treatment completion was associated with use of shorter rifamycin-based regimens. Interpreter type was not associated with LTBI treatment offer, acceptance, or completion.

## Conclusions

We found greater LTBI treatment acceptance was associated with interpreter use among non-U.S.–born individuals.

## Introduction

In 1989, the Centers for Disease Control and Prevention (CDC) established the domestic goal of tuberculosis (TB) elimination (defined as TB incidence <1 per 1 million persons) by 2010. In 2021, TB incidence in the United States was 2.37 cases per 100,000 persons, a level more than 20 times that required for TB elimination [2]. One explanation for the slow progress towards US TB elimination was a shift in the epidemiology of TB in the United States: in 1993 only 29% of TB occurred in non-U.S.–born individuals while in 2021 the frequency was 71% [1, 2]. Latent TB infection (LTBI) screening and treatment interventions that are tailored to optimize acceptance among persons who are non-U.S.–born will be essential for TB elimination in the United States.

LTBI treatment is highly effective in preventing progression of LTBI to TB disease [3]. A recent study by the Tuberculosis Epidemiologic Studies Consortium (TBESC) found that only 32% of individuals diagnosed with LTBI completed treatment [4]. Evaluating the stages of the cascade of care for LTBI treatment may identify where patient losses occur [5]. In the TBESC study, although there were losses at each care-cascade step, the biggest drop-off was seen at treatment initiation. Given high TB [2] and LTBI [6] rates in non-U.S.–born persons compared to U.S.-born populations, it is important to understand barriers to LTBI treatment and completion that are unique to non-U.S.–born persons, many of whom are not native English speakers.

Lack of English proficiency may lead to miscommunications between physicians and other providers with patients, lowers the number of healthcare visits, and generates lower patient satisfaction [7]. Across different medical settings, the use of professional interpreters has been shown to be associated with decreased communication errors, greater patient comprehension and satisfaction, and improved clinical outcomes [7]. Additionally, lack of English proficiency creates a language barrier that further promotes health disparities that exist among people who are at higher risk for LTBI and TB. There are few published studies on the effects of limited English proficiency on the LTBI treatment care cascade. A 2019 systematic review of U.S. healthcare-based strategies to improve LTBI testing and linkage to care in non-U.S.–born groups [8] identified one study that evaluated interpreter usage. In this study, language concordance between patients and providers was compared to use of a trained interpreter and found no difference in referrals for LTBI testing or receipt of testing [9]. A study in Sweden found that interpreter-assisted appointments were associated with higher rates of completion of LTBI treatment among persons seeking asylum [10].

Given the importance of addressing healthcare inequities and improving outcomes across the LTBI care continuum in non-U.S.–born persons, we investigated associations between the

use of trained medical interpreters and LTBI treatment acceptance and completion in a TBESC study as our primary outcomes. We hypothesized that the use of a trained medical interpreter during the initial interview when LTBI treatment was discussed and offered would increase LTBI treatment acceptance and completion. We also investigated whether use of a trained interpreter was associated with clinician decisions to offer treatment and whether interpreter type was associated with LTBI treatment offer, acceptance, or completion.

## Methods

### Study population and design

TBESC enrolled children and adults from July 20, 2012 to May 5, 2017, across 18 TBESC-affiliated clinics in 11 U.S. states to compare LTBI diagnostics and assess their predictive capabilities to detect progression of those with LTBI to TB disease [11]. All participants were considered to be at high risk for LTBI or progression to TB disease and included persons who were (a) close contacts of persons with infectious TB; (b) born in countries whose populations residing in the United States had high (≥100 cases/100,000 population) TB rates [12]; (c) recent arrivals (≤5 years) from countries whose populations residing in the United States had moderate (10–99 cases/100,000 population) TB rates [12]; (d) visitors of ≥30-day duration during the previous 5 years to countries whose populations had high TB rates; (e) living with HIV infection; (f) immigrants and refugees who had an abnormal chest radiograph result during the immigration process; and (g) members of a population with local LTBI prevalence of ≥25% [11]. For participants with more than one eligibility criterion, a hierarchy of enrollment reasons was established to assign a category to this variable in regression models: 1) close contact of person with infectious TB, 2) non-U.S.–born person from a high-risk country or recently arrived from medium risk country (S1 Table), 3) visitor of ≥30 days in a high-risk country during prior 5 years, 4) person belonging to a population with a LTBI prevalence ≥25%, and 5) person living with HIV.

Study staff collected blood for two FDA-approved interferon-gamma release assays (IGRAs), the QuantiFERON-TB Gold In-Tube (QFT-GIT, Qiagen Diagnostics; Hamburg, Germany) and T-SPOT.*TB* (T-SPOT, Oxford Immunotec; Oxford, UK), and placed a tuberculin skin test (TST) using the Mantoux method. Valid QFT-GIT results were defined as positive (≥0.35 IU/mL), negative, or indeterminate based on manufacturer recommendations. Study procedures allowed an indeterminate QFT test to be rerun with the same blood sample. T-SPOT results were interpreted using U.S. definitions in which negative results are defined as ≤4 spots, positive results as ≥8 spots, and borderline results as 5–7 spots. For analyses, borderline T-SPOT results were grouped into the negative category. Valid TSTs results were read within 44–76 hours of placing the test by a healthcare worker trained to read TST results, based on concerns by study sites that the recommended interval of 48–72 hours was too strict. Positive results were defined as ≥5 millimeters (mm) for high-risk persons (including close contacts and persons with HIV infection) and ≥10 mm for all other participants [13].

Participants were eligible for the current study if at least one test result for LTBI was positive. Due to participants receiving three TB infection tests during enrollment (i.e., TST, QFT-GIT, T-SPOT) and the possibility of repeat testing, criteria were developed to determine which TB infection tests would be considered for study purposes. All three TB infection tests performed within 14 days of each other for a participant were considered a "set." If a participant had more than one result from a type of TB infection test (i.e., TST, QFT-GIT, or T-SPOT) within the 14-day period, the test result performed closest to the enrollment date was used for study purposes. We excluded U.S.-born participants from analyses due to our primary research questions. Decisions to recommend LTBI treatment were at the discretion of clinic

providers. The results from decisions to offer LTBI treatment and participant acceptance and completion of treatment were reported to CDC as determined by study site clinic providers and clinic-specific practices. Due to the possibility of a participant having more than one round of LTBI treatment, the most recent and most complete round of LTBI treatment was considered the main LTBI treatment. In the case that the most recent treatment was not the most complete, the most recent regimen was preferred.

Participant-reported (or parent/legal guardian for participants ≤17 years) demographics and medical history were collected at enrollment by using standardized instruments by trained study staff. TBESC sites used the following question proposed by the U.S Census, "How well do you speak English? Would you say you speak English: Very well, Well, Not well, Not at all." Those who answered "very well" were interviewed in English unless they requested an interpreter. All other individuals were offered an interpreter in the language of their choice. The results from these language proficiency questions were not available to our study. Participants who declined an interpreter were interviewed in English. Non-U.S.–born participants who did not use an interpreter did not have their native language captured in the study database. Interpreter types were telephone-based trained interpreter, an in-person trained interpreter, or a bilingual study interviewer, based on availability and at study site discretion. Training as a medical interpreter was not required for bilingual study interviewers. Family members, friends or other patients could not be interpreters. Treatment regimen categories included daily isoniazid for 6 or 9 months, daily rifampin for 4 months, weekly isoniazid/rifapentine for 12 weeks, and all other regimens. Participants could indicate one or more racial/ethnic categories. For participants who chose Hispanic/Latino and any other racial category, we designated their race/ethnicity as Hispanic/Latino. Other combinations of racial categories were included in the "Other" category due to small sample sizes.

## Statistical analysis

Descriptive statistics were used to examine participants lost to follow-up, enrollment variations by clinic, and participants included in the study. Chi square or Fisher's statistics were used to compare groups. Our primary outcomes of interest were participants' acceptance of LTBI treatment and successful completion of LTBI treatment. Our primary predictor of interest was whether the use of an interpreter was associated with the outcomes of interest. We also evaluated whether interpreter usage was associated with a decision to offer LTBI treatment and whether the type of interpreter was associated with study outcomes.

We assessed for associations between our predictors of interest and each of the three LTBI treatment outcomes (offered, accepted, and completed LTBI treatment) using multivariable logistic regression models with random intercepts (*melogit* command in Stata) to account for TBESC site clustering. The following covariates were assessed: use of an interpreter, age, gender, race/ethnicity, enrollment indication, time residing in the United States, World Health Organization (WHO) region of birth, level of education, housing status, HIV status, diabetes, TB infection test results (positive or negative), and LTBI treatment regimen. For each model, if a variable had >10% missing information (including "don't know/refused"), then the variable was dropped, including T-SPOT results, Bacille Calmette-Guérin (BCG) vaccine status, refugee status, income, injection drug use, alcohol consumption, correctional facility, holding center, and long-term care facility. We assessed for multicollinearity between our independent variables using variance inflation factors and condition indices. A two-sided $P$-value ≤0.05 was considered significant. All statistical analyses were performed using StataSE 17 (StataCorp, College Station, TX).

### Ethics approvals

All participants provided written informed consent. The study was approved by the CDC's institutional review board (IRB) and the IRBs of Johns Hopkins University School of Medicine, University of Maryland, Maryland Department of Health, North Texas Regional, and Atrium Health. Study authors (YS, MN) had access to identifiable data from a single study site during the data collection period.

## Results

TBESC enrolled 22,131 participants at 18 TBESC-affiliated sites, of whom 9,531 participants had at least one positive TB infection test result and no evidence of active TB (Fig 1). Of the 9,531 participants diagnosed with LTBI, 8,761 were non-U.S.–born (91.9%) and included in our analyses, among whom 6,272 (71.6%) used an interpreter and 2,489 (28.4%) had their interview conducted in English (Table 1). Interpreter use varied by enrollment site with the highest at DeKalb County, Georgia (26.0%) and the lowest at Florida Department of Health—Gainesville (0.1%) and Montgomery County in Maryland (0.1%) (S2 Table). Among participants who used an interpreter, 2,248 (35.8%) used an in-person interpreter, 1,873 (29.9%) used a telephonic interpreter, and 2,151 (34.3%) used a bilingual member of the study staff (S3 Table). The five most common countries of origin were Myanmar (1,492, 17.0%), the Philippines (1,206, 13.8%), Bhutan (806, 9.2%), Mexico (522, 6.0%), and Somalia (388, 4.4%) (Table 1). Among participants who used an interpreter, the three most frequent languages were Burmese (n = 1,405, 22.4%), Spanish (n = 1,123, 17.9%) and Nepali (1,087, 17.3%). From the time that non-U.S.–born participants were screened for LTBI to completion of treatment, we found that there were losses at each stage of the cascade: 4,158 of 8,761 were offered LTBI treatment (47.5%), 3,789 of 4158 accepted treatment (91.1%) and 2,990 of 3789 completed treatment (78.9%) (Fig 1). Among non-U.S.–born participants with at least one positive LTBI test result, 2990 out of 8761 (34.1%) completed LTBI treatment.

There was no association between use of an interpreter and a clinician's decision to offer LTBI treatment in a multivariable model (S4 Table). Compared to participants whose enrollment indication was contact investigation, all other enrollment indications had lower odds of being offered treatment. In comparison to Asian participants, participants who identified as Pacific Islander persons had lower odds of being offered treatment (adjusted odds ratio [aOR] 0.64, 95% confidence interval [CI] 0.41–1.0). Participants who were experiencing homelessness also had lower odds of being offered treatment (aOR 0.41, 95% CI 0.26–0.65). Additional characteristics associated with decreased odds of treatment offer included longer time since U.S. entry and birthplace in the European and Pacific regions.

Participants who used an interpreter had greater odds of accepting LTBI treatment (aOR 1.66, 95% CI 1.18–2.33) (Table 2). Factors associated with lower odds of accepting LTBI treatment included Black race (aOR 0.54, 95% CI 0.31–0.95), Hispanic/Latino ethnicity (aOR 0.31, 95% CI 0.13–0.73) or "other" race (aOR 0.65, 95% CI 0.42–1.00), having diabetes (aOR 0.62, 95% CI 0.39–0.98), and attaining postgraduate-level education (aOR 0.33, 95% CI 0.16–0.69). Compared to the African region, participants born in the European and Pacific regions had lower odds of accepting treatment (aOR 0.18, 95% CI 0.06–0.54 and aOR 0.37, 95% CI 0.18–0.74, respectively). Participants living with HIV had greater odds of accepting treatment (aOR 8.03, 95% CI 0.93–69.66). Accepted treatment regimens were 4 months of rifampin (22.2%), 12 weeks of isoniazid/rifapentine (5.5%), 6 or 9 months of isoniazid (13.1%) and "Other" treatments (2.4%).

Interpreter usage was not associated with greater treatment completion (aOR 1.29, 95% CI 0.98–1.70) (Table 3). Participants who were born in the Mediterranean WHO region had lesser

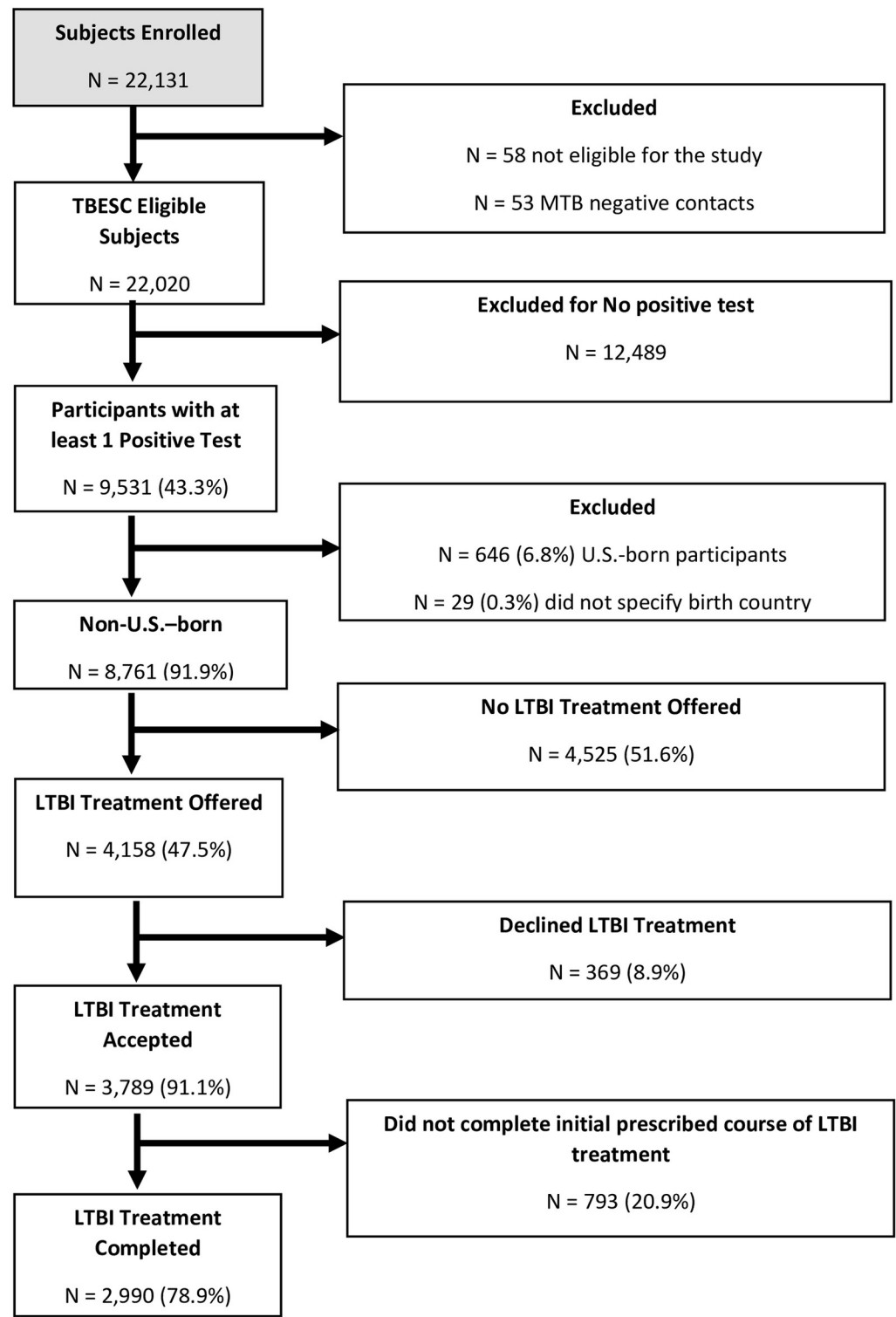

**Fig 1. Participants enrolled from July 20, 2012, to May 5, 2017.**

**Table 1. Characteristics of non-U.S.–born Tuberculosis Epidemiologic Studies Consortium participants by interview language during initial interview to determine eligibility for treatment of latent tuberculosis infection.**

| Characteristic | Non-U.S.–born persons | | | | | |
|---|---|---|---|---|---|---|
| | Total N = 8,761 (100%) | | Interview in English N = 2,489 (28.4%) | | Interview in language other than English N = 6,272 (71.6%) | |
| Age in years<br>Mean age: 35.68<br>Range: [1, 97] | No. | % | No. | % | No. | % |
| 0–14 | 1,015 | 11.6% | 209 | 8.4% | 806 | 12.9 |
| 15–24 | 1,420 | 16.2% | 454 | 18.2% | 966 | 15.4% |
| 25–44 | 3,990 | 45.5% | 1,067 | 42.9% | 2,923 | 46.6% |
| 45–64 | 1,953 | 22.3% | 664 | 26.7% | 1,289 | 20.6% |
| ≥65 | 383 | 4.4% | 95 | 3.8% | 288 | 4.6% |
| Gender | | | | | | |
| Women | 4,122 | 47.1% | 1,188 | 47.7% | 2,934 | 46.8% |
| Men | 4,636 | 52.9% | 1,300 | 52.2% | 3,336 | 53.2% |
| Transgender[1] | 3 | 0.0% | 1 | 0.0% | 2 | 0.0% |
| Race/Ethnicity | | | | | | |
| Asian | 3,311 | 37.8% | 982 | 39.5% | 2,329 | 37.1% |
| Black/African American | 1,333 | 15.2% | 433 | 17.4% | 900 | 14.4% |
| Hispanic/Latino | 1,053 | 12.0% | 202 | 8.1% | 851 | 13.6% |
| White | 318 | 3.6% | 104 | 4.2% | 214 | 3.4% |
| Pacific Islander | 174 | 2.0% | 143 | 5.8% | 31 | 0.5% |
| Native American[2] | 4 | 0.1% | 2 | 0.1% | 2 | 0.0% |
| Other | 2,047 | 23.4% | 482 | 19.4% | 1,565 | 25.0% |
| Unknown | 521 | 6.0% | 141 | 5.7% | 380 | 6.1% |
| Enrollment Reason | | | | | | |
| Close contact | 708 | 8.1% | 365 | 14.7% | 343 | 5.5% |
| Non-U.S.–born | 7,632 | 87.1% | 2,012 | 80.8% | 5,620 | 89.6% |
| Member of a group with local LTBI prevalence ≥25%[3] | 358 | 4.1% | 69 | 2.8% | 289 | 4.6% |
| Spent at least 30 days in a high-risk country in the last 5 years[4] | 27 | 0.3% | 22 | 0.9% | 5 | 0.1% |
| HIV positive | 36 | 0.4% | 21 | 0.8% | 15 | 0.2% |
| Time since arrival to the US | | | | | | |
| Years (med, IQR) | 8,719 | 0.2 (0.1–1.5) | 2,473 | 0.62 (0.1–8.4) | 6,246 | 0.1 (0.1–0.3) |
| <5 years | 7,120 | 81.3% | 1,667 | 67.0% | 5,453 | 81.3% |
| ≥5 years | 1,641 | 18.7% | 822 | 33.0% | 819 | 13.1% |
| HIV infection | | | | | | |
| Yes | 92 | 1.1% | 43 | 1.7% | 54 | 0.9% |
| No | 8,601 | 98.2% | 2,432 | 97.7% | 6,169 | 98.4% |
| Don't know/refused | 68 | 0.8% | 27 | 0.9% | 54 | 0.9% |
| Diabetes mellitus | | | | | | |
| Yes | 424 | 4.8% | 161 | 6.5% | 263 | 4.2% |
| No | 8,270 | 94.4% | 2,308 | 92.7% | 5,962 | 95.1% |
| Don't Know/refused | 67 | 0.8% | 20 | 0.8% | 47 | 0.8% |
| Experiencing homelessness | | | | | | |
| Yes | 142 | 1.6% | 66 | 2.7% | 76 | 1.2% |
| No | 8598 | 98.1% | 2415 | 97.0% | 6183 | 98.6% |
| Don't know/refused | 21 | 0.2% | 8 | 0.3% | 13 | 0.2% |
| Injection drug use (n = 217) | | | | | | |

*(Continued)*

**Table 1.** (Continued)

| Characteristic | Non-U.S.–born persons | | | | | |
|---|---|---|---|---|---|---|
| | Total N = 8,761 (100%) | | Interview in English N = 2,489 (28.4%) | | Interview in language other than English N = 6,272 (71.6%) | |
| Age in years Mean age: 35.68 Range: [1, 97] | No. | % | No. | % | No. | % |
| Yes | 6 | 0.1% | 4 | 0.2% | 2 | 0.0% |
| No | 210 | 2.4% | 101 | 4.1% | 109 | 1.7% |
| Don't know/refused | 1 | 0.0% | 0 | 0.0% | 1 | 0.0% |
| Consumption of 4 or more drinks containing alcohol (n = 2655) | | | | | | |
| Never | 1,546 | 17.6% | 625 | 25.1% | 921 | 14.7% |
| Once a month or less | 733 | 8.4% | 268 | 10.8% | 465 | 7.4% |
| 2–3 times a month | 146 | 1.7% | 57 | 2.3% | 89 | 1.4% |
| Once per week | 102 | 1.2% | 28 | 1.1% | 74 | 1.2% |
| 2–3 times a week | 57 | 0.7% | 15 | 0.6% | 42 | 0.7% |
| 4 or more times a week | 26 | 0.3% | 6 | 0.2% | 20 | 0.3% |
| Don't Know/refused | 45 | 0.5% | 15 | 0.6% | 30 | 0.5% |
| Correctional facility[5] | | | | | | |
| Yes | 312 | 3.6% | 64 | 2.6% | 248 | 4.0% |
| No | 7738 | 88.3% | 2283 | 91.7% | 5455 | 87.0% |
| Don't know/refused | 9 | 0.1% | 4 | 0.2% | 5 | 0.1% |
| Missing | 702 | 8.0% | 138 | 5.5% | 546 | 9.0% |
| Holding Center[6] | | | | | | |
| Yes | 2,381 | 27.2% | 141 | 5.7% | 2,240 | 35.7% |
| No | 6335 | 72.3% | 2321 | 93.3% | 4014 | 64.0% |
| Don't know/refused | 45 | 0.5% | 27 | 1.1% | 18 | 0.3% |
| Long-term care facility | | | | | | |
| Yes | 307 | 3.5% | 190 | 7.6% | 117 | 1.9% |
| No | 8440 | 96.3% | 2288 | 91.9% | 6152 | 98.1% |
| Don't know/refused | 14 | 0.2% | 11 | 0.4% | 3 | 0.1% |
| Country of birth (5 most common) | | | | | | |
| Myanmar | 1,492 | 17.0% | 23 | 0.9% | 1,469 | 23.4% |
| Philippines | 1,206 | 13.8% | 967 | 38.9% | 239 | 3.8% |
| Bhutan | 806 | 9.2% | 12 | 0.5% | 794 | 12.7% |
| Mexico | 522 | 6.0% | 100 | 4.0% | 422 | 6.7% |
| Somalia | 388 | 4.4% | 46 | 1.9% | 342 | 5.5% |
| Region of birth country | | | | | | |
| Africa | 906 | 10.3% | 259 | 10.4% | 647 | 10.3% |
| America | 1,730 | 19.8% | 442 | 17.8% | 1,288 | 20.5% |
| Europe | 98 | 1.1% | 46 | 1.9% | 52 | 0.8% |
| Mediterranean | 1,049 | 12.0% | 192 | 7.7% | 857 | 13.7% |
| Pacific | 2,154 | 24.6% | 1,357 | 54.5% | 797 | 12.7% |
| Southeast Asia | 2,824 | 32.3% | 193 | 7.8% | 2,631 | 42.0% |
| Education | | | | | | |
| No schooling | 1,046 | 11.9% | 25 | 1.0% | 1,021 | 16.3% |
| Eighth grade or less | 2,642 | 30.2% | 151 | 6.1% | 2,491 | 39.7% |
| Some high school | 1,270 | 14.5% | 242 | 9.7% | 1,028 | 16.4% |
| High school graduate or GED | 1,604 | 18.3% | 567 | 22.8% | 1,037 | 16.5% |

(*Continued*)

**Table 1.** (Continued)

| Characteristic | Non-U.S.–born persons | | | | | |
|---|---|---|---|---|---|---|
| | Total N = 8,761 (100%) | | Interview in English N = 2,489 (28.4%) | | Interview in language other than English N = 6,272 (71.6%) | |
| Age in years<br>Mean age: 35.68<br>Range: [1, 97] | No. | % | No. | % | No. | % |
| Trade school or associates degree | 186 | 2.1% | 126 | 5.1% | 60 | 1.0% |
| Some university/college | 730 | 8.3% | 508 | 20.4% | 222 | 3.5% |
| University/college graduate | 1,005 | 11.5% | 659 | 26.5% | 346 | 5.5% |
| Postgraduate schooling | 239 | 2.7% | 202 | 8.1% | 37 | 0.6% |
| Other | 7 | 0.1% | 0 | 0.0% | 7 | 0.1% |
| Don't know/refused | 32 | 0.4% | 9 | 0.4% | 23 | 0.4% |
| LTBI Treatment regimen offered | | | | | | |
| 6- or 9- months isoniazid | 700 | 8.0% | 232 | 9.3% | 468 | 7.5% |
| 4 months rifampin | 1,697 | 19.4% | 382 | 15.3% | 1,315 | 21.0% |
| 12 weeks- weekly doses isoniazid/rifapentine | 885 | 10.1% | 166 | 6.7% | 719 | 11.5% |
| Other[7] | 875 | 10.0% | 108 | 4.3% | 767 | 12.2% |
| LTBI Treatment regimen received | | | | | | |
| 6- or 9-months isoniazid | 1,152 | 13.1% | 249 | 10.0% | 903 | 14.4% |
| 4 months rifampin | 1,943 | 22.2% | 319 | 12.8% | 1,624 | 25.9% |
| 12 weeks- weekly doses isoniazid/rifapentine | 484 | 5.5% | 115 | 4.6% | 369 | 5.9% |
| Other | 210 | 2.4% | 71 | 2.9% | 139 | 2.2% |
| Offered LTBI treatment | | | | | | |
| Yes | 4,158 | 47.5% | 889 | 35.7% | 3,269 | 52.1% |
| No | 4,525 | 51.6% | 1,571 | 63.1% | 2,954 | 47.1% |
| Missing | 78 | 0.9% | 29 | 1.2% | 49 | 0.8% |
| Accepted LTBI treatment | | | | | | |
| Yes | 3,789 | 91.1% | 754 | 30.3% | 3,035 | 48.4% |
| No | 369 | 8.9% | 135 | 5.4% | 234 | 3.7% |
| Missing | 0 | 0.0% | 0 | 0.0% | 0 | 0.0% |
| Completed LTBI treatment | | | | | | |
| Yes | 2,990 | 78.9% | 575 | 23.1% | 2,415 | 38.5% |
| No | 793 | 20.9% | 177 | 7.1% | 616 | 9.8% |
| Missing | 6 | 0.2% | 2 | 0.1% | 4 | 0.1% |
| Tuberculin Skin Test[8] | | | | | | |
| Positive | 7,822 | 89.3% | 2,179 | 87.5% | 5,643 | 90.0% |
| Negative | 864 | 9.9% | 284 | 11.4% | 580 | 9.2% |
| QuantiFERON-TB Gold In-Tube | | | | | | |
| Positive | 4,634 | 52.9% | 1,297 | 52.1% | 3,337 | 53.2% |
| Negative | 4,052 | 46.3% | 1,169 | 47.0% | 2,883 | 46.0% |
| T-SPOT.*TB* Test | | | | | | |
| Positive | 3,610 | 41.5% | 932 | 37.9% | 2,678 | 43.0% |
| Negative | 4,128 | 46.8% | 1,177 | 46.9% | 2,951 | 46.8% |

(*Continued*)

**Table 1.** (Continued)

| Characteristic | Non-U.S.–born persons | | | | | |
|---|---|---|---|---|---|---|
| | Total<br>N = 8,761 (100%) | | Interview in English<br>N = 2,489 (28.4%) | | Interview in language other than English<br>N = 6,272 (71.6%) | |
| Age in years<br>Mean age: 35.68<br>Range: [1, 97] | No. | % | No. | % | No. | % |
| Borderline | 547 | 6.2% | 152 | 6.1% | 395 | 6.3% |

[1]Transgender participants were dropped from models due to other missing values

[2]Native American participants were dropped from models due to missing values

[3]Populations with a prevalence of LTBI ≥ 25% varied by site (e.g., individuals experiencing homelessness or have a specific medical condition)

[4]Refer to supplemental S1 Table for a list of high-risk countries

[5]Correctional facility such as prison or jail

[6]Holding center such as refugee camp or refugee detention

[7]Other regimens included: Ethambutol, Pyrazinamide, Levofloxacin, Moxifloxacin, Isoniazid/Rifampin

[8]Tuberculin skin test measured in millimeters of induration with positivity determined by LTBI risk (see Methods)

odds of completing treatment (aOR 0.49, 95% CI 0.35–0.70) compared to participants born in the African region. Participants who were experiencing homelessness had lower odds of completing treatment (aOR 0.49, 95% CI 0.24–0.99). Compared to treatment with 6 or 9 months of isoniazid, participants treated with all other regimens had greater odds of completing treatment.

We evaluated the effects of interpreter type on treatment outcomes. All interpreter types were trained in medical interpretation and included telephone-based, in-person, or bilingual study interviewers (S4 Table). Interpreter type was not associated with a decision to offer, accept, or complete LTBI treatment (S5–S7 Tables, **respectively**).

## Discussion

In a large multisite study, we evaluated the effect of the use of interpreters on LTBI treatment offer, acceptance, and completion in non-U.S.–born persons. We found that the use of a trained medical interpreter was associated with a 66% increased odds of treatment acceptance. There was no statistically significant association between interpreter use and a decision to offer treatment or treatment completion. Many non-U.S.–born persons are at increased risk for both LTBI and progression to TB. Among non-U.S.–born persons who have limited English proficiency, their understanding of LTBI might be affected by whether they receive patient education about TB infection in English or in their preferred language through the use of an interpreter.

Consistent with prior research [4], we found that the largest losses in the LTBI treatment continuum (over 50%) occurred at the step of offering treatment, which was at clinicians' discretion.. Participants who were enrolled on the basis of non-U.S. birth, HIV infection or local LTBI prevalence of at least 25% had lesser odds of being offered LTBI treatment than close contacts to a TB case. In order to improve LTBI care continuum outcomes, reasons for these differences should be investigated. Possibly reflecting health disparities, participants who identified as Pacific Islander persons and persons experiencing homelessness had lower odds of being offered LTBI treatment. For treatment acceptance, it is concerning that persons whose race/ethnicities were Black, Latino, or "other", and those with diabetes were less likely to accept

**Table 2. Adjusted odds ratios for acceptance of LTBI treatment by use of an interpreter.** N = 3,973.

| Characteristics | Adjusted odds Ratio | 95% confidence interval | p-value |
|---|---|---|---|
| **Interpreter** | 1.66 | 1.18–2.33 | 0.004 |
| **Time in the US (years)** | 1.02 | 1.00–1.05 | 0.07 |
| **Gender** | | | |
| Women | reference | | |
| Men | 0.98 | 0.77–1.26 | 0.90 |
| **Enrollment reason[1]** | | | |
| Close contact | reference | | |
| Non-U.S.–born | 0.61 | 0.36–1.03 | 0.06 |
| Member of group with local LTBI prevalence[2] ≥25% | 0.60 | 0.17–2.09 | 0.42 |
| HIV infection | 0.36 | 0.01–8.69 | 0.53 |
| **Age (Years)** | 0.98 | 0.97–0.99 | <0.001 |
| **Race/ethnicity** | | | |
| Asian | reference | | |
| Black/African American | 0.54 | 0.31–0.95 | 0.03 |
| Hispanic/Latino | 0.31 | 0.13–0.73 | 0.007 |
| White | 0.60 | 0.30–1.17 | 0.13 |
| Pacific Islander | 6.79 | 0.88–52.62 | 0.07 |
| Other | 0.65 | 0.42–1.00 | 0.05 |
| Unknown | 1.02 | 0.47–2.24 | 0.96 |
| **Region of birth country** | | | |
| Africa | reference | | |
| America | 1.21 | 0.56–2.61 | 0.63 |
| Europe | 0.18 | 0.06–0.54 | 0.002 |
| Mediterranean | 0.65 | 0.41–1.02 | 0.06 |
| Pacific | 0.37 | 0.18–0.74 | 0.005 |
| Southeast Asia | 0.91 | 0.51–1.64 | 0.77 |
| **Education** | | | |
| No schooling | reference | | |
| Eighth grade or less | 1.18 | 0.77–1.81 | 0.44 |
| Some high school | 1.02 | 0.62–1.67 | 0.94 |
| High school graduate or GED | 0.94 | 0.58–1.53 | 0.81 |
| Trade school or associates degree | 0.87 | 0.38–1.98 | 0.71 |
| Some university/college | 0.74 | 0.41–1.32 | 0.31 |
| University/college graduate | 0.68 | 0.41–1.15 | 0.15 |
| Postgraduate schooling | 0.33 | 0.16–0.69 | 0.003 |
| **Housing Status** | 3.00 | 0.60–15.02 | 0.18 |
| Housed | reference | | |
| Experiencing homelessness | 3.00 | 0.60–15.02 | 0.18 |
| **HIV** | | | |
| HIV uninfected | reference | | |
| Living with HIV infection | 8.03 | 0.93–69.66 | 0.06 |
| **Diabetes** | | | |
| Without diabetes | reference | | |
| Living with diabetes | 0.62 | 0.39–0.98 | 0.04 |
| **LTBI treatment accepted** | | | |
| 6- or 9- months isoniazid | reference | | |
| 4 months rifampin | 1.21 | 0.74–1.96 | 0.45 |

(*Continued*)

**Table 2.** (Continued)

| Characteristics | Adjusted odds Ratio | 95% confidence interval | p-value |
|---|---|---|---|
| 12 weeks- weekly doses isoniazid/rifapentine | 1.61 | 0.97–2.69 | 0.07 |
| Other[3] | 0.90 | 0.48–1.71 | 0.75 |
| **Tuberculin Skin Test (TST)** | | | |
| TST Negative | reference | | |
| TST Positive | 0.99 | 0.68–1.44 | 0.95 |
| **QuantiFERON-TB Gold In-Tube[4]** | | | |
| Negative QuantiFERON-TB | reference | | |
| Positive QuantiFERON-TB | 1.35 | 0.97–1.88 | 0.08 |

[1]The variable "Spent at least 30 days in a high-risk country in the last 5 years" was dropped from the model due to other missing values

[2]Populations with a prevalence of LTBI ≥ 25% varied by site (e.g., individuals experiencing homelessness or have a specific medical condition)

[3]Other regimens included: Ethambutol, Pyrazinamide, Levofloxacin, Moxifloxacin, Isoniazid/Rifampin

[4]T-SPOT.TB test excluded due to >10% missing data

LTBI treatment. The finding that higher levels of education were associated with lower treatment acceptance was unexpected. We hypothesize that this may be due to misconceptions about test results in the setting of BCG vaccination or about TB occurrence in persons with higher socioeconomic status. Lack of housing was independently associated with lower likelihood of treatment completion, underscoring the need to develop interventions specific to persons who experience homelessness. Further investigations into these associations are warranted. Similar to other studies [14–16], shorter course rifamycin-based LTBI treatment regimens are associated with decreased losses across the care continuum.

There was no association between the type of interpreter and LTBI outcomes. All three interpreter types were effective in increasing treatment acceptance. In-person interpreters may not be available at all clinical sites of practice, and our findings should provide reassurance around the effect of telephone-based interpretation. Although bilingual study interviewers are a type that is unique to study settings, there may be similarities to bilingual clinicians, and this deserves further investigation.

There are several limitations to this study. TBESC sites used a standard question to assess English proficiency and offered interpreters on the basis of the response. Participants who answered less proficient than "very well" could decline use of an interpreter and conduct the interview in English. As we did not have access to the responses to English proficiency, we could not evaluate differences in interpreter impact by objective measures of English proficiency. We were unable to further explore differences in LTBI outcomes by language as the language of participants were collected only for those who accepted an interpreter. A trained interpreter may not have been available for the preferred language of participants, resulting in a situation in which participants may have still been disadvantaged due to limited language proficiency. Observed associations between variables and our outcomes of interest may have been driven by site-specific differences. To address this, we used random intercept models to adjust for enrollment site. Our study did not address earlier steps in the LTBI treatment cascade, such as a decision to test and communication of the results to patients, where the greatest losses may occur [17].

In summary, our study found that non-U.S.–born people with limited English proficiency may benefit from the use of an interpreter regardless of the interpreter method (bilingual interpreter, in-person, by telephone). We identified a number of possible health inequities

**Table 3. Adjusted odds ratios for completion of LTBI treatment by use of an interpreter.** [1] N = 3,626.

| Characteristics | Adjusted Odds Ratio | 95% Confidence Interval | p-value |
|---|---|---|---|
| **Interpreter** | 1.28 | 0.97–1.69 | 0.08 |
| **Time in the US (years)** | 1.01 | 0.99–1.03 | 0.42 |
| **Gender** | | | |
| Women | reference | | |
| Men | 1.14 | 0.95–1.35 | 0.15 |
| **Enrollment reason**[2] | | | |
| Close contact | reference | | |
| Non-U.S.–born | 0.94 | 0.66–1.33 | 0.72 |
| Member of group with local LTBI prevalence ≥25%[3] | 1.30 | 0.59–2.83 | 0.51 |
| HIV positive | 0.52 | 0.13–2.13 | 0.37 |
| **Age (years)** | 1.00 | 0.99–1.00 | 0.21 |
| **Race/ethnicity** | | | |
| Asian | reference | | |
| Black/African American | 0.76 | 0.51–1.14 | 0.19 |
| Hispanic/Latino | 0.71 | 0.41–1.22 | 0.21 |
| White | 0.64 | 0.38–1.08 | 0.10 |
| Pacific Islander | 0.56 | 0.20–1.59 | 0.28 |
| Other | 0.91 | 0.69–1.21 | 0.52 |
| Unknown | 0.82 | 0.52–1.30 | 0.40 |
| **Region of birth country** | | | |
| Africa | reference | | |
| America | 0.61 | 0.36–1.04 | 0.07 |
| Europe | 0.33 | 0.09–1.21 | 0.10 |
| Mediterranean | 0.49 | 0.35–0.70 | <0.001 |
| Pacific | 0.70 | 0.41–1.21 | 0.20 |
| Southeast Asia | 1.05 | 0.69–1.62 | 0.81 |
| **Education**[4] | | | |
| No schooling | reference | | |
| Eighth grade or less | 1.02 | 0.78–1.35 | 0.87 |
| Some high school | 1.22 | 0.88–1.69 | 0.24 |
| High school graduate or GED | 1.34 | 0.96–1.87 | 0.09 |
| Trade school or associate degree | 1.58 | 0.81–3.08 | 0.18 |
| Some university/college | 1.49 | 0.95–2.33 | 0.08 |
| University/college graduate | 1.52 | 1.01–2.30 | 0.04 |
| Postgraduate schooling | 1.92 | 0.81–4.52 | 0.14 |
| Don't know/refused | 0.46 | 0.14–1.47 | 0.19 |
| **Housing status** | | | |
| Housed | reference | | |
| Experiencing homelessness | 0.49 | 0.24–0.98 | 0.045 |
| **HIV** | | | |
| HIV uninfected | reference | | |
| Living with HIV infection | 1.49 | 0.62–3.60 | 0.37 |
| **Diabetes** | | | |
| Without diabetes | reference | | |
| Living with diabetes | 1.04 | 0.69–1.57 | 0.85 |
| **LTBI treatment initially accepted** | | | |
| 6- or 9-months isoniazid | reference | | |

*(Continued)*

**Table 3.** (Continued)

| Characteristics | Adjusted Odds Ratio | 95% Confidence Interval | p-value |
|---|---|---|---|
| 4 months rifampin | 1.34 | 1.07–1.70 | 0.01 |
| 12 weeks- weekly doses isoniazid/rifapentine | 2.77 | 1.95–3.93 | <0.001 |
| Other regimens | 1.73 | 1.11–2.70 | 0.02 |
| **Tuberculin Skin Test** | 1.21 | 0.92–1.60 | 0.17 |
| **QuantiFERON-TB Gold In-Tube** | 1.15 | 0.92–1.44 | 0.21 |

[1]T-SPOT.TB test excluded due to >10% missing data

[2] Participants enrolled due to having spent at least 30 days in a high-risk country were removed from this model due to a small number (n = 6)

[3]Populations with a prevalence of LTBI ≥ 25% varied by site (e.g., individuals experiencing homelessness or have a specific medical condition)

[4]Participants enrolled with an education level of "other" were removed from this model due to a small number (n = 4)

associated with ethnicity and housing status that should be investigated further. Finally, clinician offering of LTBI treatment appeared to be suboptimal in our study, regardless of whether an interpreter was used. Use of an interpreter, in addition to shorter course regimens for LTBI treatment, increases treatment completion rates among non-U.S.–born persons and is an important intervention for addressing health disparities among persons with limited English proficiency.

## Tuberculosis Epidemiologic Studies Consortium

*California Department of Public Health*, *Richmond* Jennifer Flood, Lisa Pascopella (includes San Francisco Department of Public Health, Julie Higashi; County of San Diego Health and Human Services Agency, Kathleen Moser, Marisa Moore [CDC]; and University of California San Diego Antiviral Research Center, Richard Garfein, Constance Benson); *Denver (CO) Health and Hospital Authority* Robert Belknap, Randall Reves; *Duke University (Durham, NC)* Jason Stout (includes Carolinas Medical Center [Charlotte, NC], Amina Ahmed; Vanderbilt University Medical Center [Nashville, TN], Timothy Sterling, April Pettit; Wake County Human Services [Raleigh, NC], Jason Stout); *Emory University (Atlanta)* Henry M Blumberg (includes DeKalb County Board of Health, Alawode Oladele); *University of Florida (Gainesville)* Michael Lauzardo, Marie Nancy Séraphin; *Hawaii Department of Health (Honolulu)* Richard Brostrom; *Maricopa County Department of Public Health (Phoenix, AZ)* Renuka Khurana; *Maryland Department of Health (Baltimore),* Wendy Cronin, Susan Dorman; *Public Health—Seattle and King County* Masahiro Narita, David Horne; *University of North Texas Health Science Center (Fort Worth)* Thaddeus Miller.

## Supporting information

**S1 Table. Countries with high rates of tuberculosis (>100 cases per 100,000 population).**
(DOCX)

**S2 Table. The number of non-U.S.–born participants enrolled at different sites by interview language, English or other than English.**
(DOCX)

**S3 Table. Characteristics of TBESC participants who used an interpreter by interpreter type.**
(DOCX)

**S4 Table. Adjusted odds ratios for the offer of LTBI treatment by use of an interpreter.**
N = 8,402.
(DOCX)

**S5 Table. Adjusted* odds ratios for the offer of LTBI treatment by interpreter type.**
N = 6,027.
(DOCX)

**S6 Table. Adjusted* odds ratios for acceptance of LTBI treatment by interpreter type.**
N = 3,114.
(DOCX)

**S7 Table. Adjusted* odds ratio for completion of LTBI treatment by interpreter type.**
N = 2,913.
(DOCX)

## Acknowledgments

The findings and conclusions in this report are those of the authors and do not necessarily represent official CDC positions.

We are grateful for the assistance of the CDC headquarters TBESC team: Gabrielle Fanning-Dowdell, Thara Venkatappa, Rose Punnoose, Matthew Whipple, Kathryn Winglee, and Yanjue Wu. We also acknowledge the contributions of the previous TBESC project officer, Denise Garrett, and branch chief, Tom Navin. We acknowledge the assistance of the TBESC site project coordinators: Katya Salcedo, Laura Romo, Christine Kozik, Carlos Vera, Juanita Lovato, Laura Farrow, Colleen Traverse Kristian Atchley, Fernanda Maruri, Kursten Lyon, Debra Turner, Nubia Flores, Jane Tapia, Livia Sura, Joanne C Li, Marie McMillan, Stephanie Reynolds-Bigby, Angela Largen, Aurimar Ayala, Elizabeth Munk, Gina Maltas, Yoseph Sorri, Kenji Matsumoto, Amy Board, and James Akkidas.

## Author Contributions

**Conceptualization:** Dolly Katz, Lauren Lambert, David J. Horne.

**Data curation:** Dolly Katz, Lauren Lambert, Yoseph Sorri, Masahiro Narita.

**Formal analysis:** Rebeca Gonzalez-Reyes, David J. Horne.

**Investigation:** Rebeca Gonzalez-Reyes.

**Methodology:** Dolly Katz, Lauren Lambert, David J. Horne.

**Project administration:** Masahiro Narita, David J. Horne.

**Resources:** Yoseph Sorri.

**Supervision:** Dolly Katz, David J. Horne.

**Writing – original draft:** Rebeca Gonzalez-Reyes, David J. Horne.

**Writing – review & editing:** Rebeca Gonzalez-Reyes, Dolly Katz, Lauren Lambert, Yoseph Sorri, Masahiro Narita, David J. Horne.

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
