## [Decision Letter · Decision Letter 0]

26 Dec 2023

PONE-D-23-18821Interpreter usage and associations with latent tuberculosis infection treatment outcomes in the USA among non-U.S.–born persons, 2012–2017PLOS ONE

Dear Dr. Horne,

Thank you for submitting your manuscript to PLOS ONE. After careful consideration, we feel that it has merit but does not fully meet PLOS ONE’s publication criteria as it currently stands. Therefore, we invite you to submit a revised version of the manuscript that addresses the points raised during the review process.

We look forward to receiving your revised manuscript.

Kind regards,

Lisa Kawatsu, PhD

Academic Editor

PLOS ONE

Journal Requirements:

Reviewers' comments:

Reviewer's Responses to Questions

**Comments to the Author**

1. Is the manuscript technically sound, and do the data support the conclusions?

Reviewer #1: Partly

Reviewer #2: Yes

2. Has the statistical analysis been performed appropriately and rigorously? 

Reviewer #1: No

Reviewer #2: Yes

3. Have the authors made all data underlying the findings in their manuscript fully available?

Reviewer #1: Yes

Reviewer #2: Yes

4. Is the manuscript presented in an intelligible fashion and written in standard English?

Reviewer #1: Yes

Reviewer #2: Yes

5. Review Comments to the Author

Reviewer #1: The article presents the evaluation of the use of an interpreter and its effect on LTBI treatment offer, acceptance and completion in non-US born individuals. The study uses a subpopulation of non-U.S.–born who participated in a previously published study that assessed LTBI diagnostics and their predictive capabilities to detect progression of those with LTBI to TB disease. This is a very important topic and the manuscript presents results that are of interest to the scientific community, however, the authors must first address a number of issues.

1. Major: The tittle refers to "treatment outcomes" however, I think this tittle may be misleading because the study does not address LTBI treatment outcomes as they may be understood e.g., disease Vs no disease or cured Vs no cured. As the authors mention in line 296-297 of the discussion section "effect of the use of interpreters on LTBI treatment offer, acceptance, and completion" that's what they evaluated. The tittle should be modified accordingly. This should also be modified in line 30 of the abstract.

2. Minor: abstract lines 34-38, it is not clear what is a primary outcome and a secondary outcome.

3. Minor: line 79 of the introduction, please include a reference to support the statement "LTBI treatment is highly effective in preventing progression of LTBI to TB disease"

4. Major: lines 106-108, Clarify what the primary and secondary outcomes are: Primary outcome seems to be treatment acceptance and completion. Secondary outcomes: association with clinicians decision to offer treatment? Association with treatment outcomes? If that's the primary outcome, as mentioned in point 1, the title of the manuscript should be changed "associations with latent tuberculosis infection treatment acceptance and completion" In lines 181-182 and 184-185 mentions that the primary outcome was acceptance and completion.

5. Minor: line 109, replace "type associated with" for "type was associated with"

6. Minor: line 125-126, why was a hierarchy of enrollment reasons necessary? There's no further mention to this or enough explanation on why it was important/needed.

7. Minor: lines 149-152 ”If a participant had more 150 than one TST, QFT-GIT, or T-SPOT test within 14 days, the TB infection test(s) where all three tests were performed closest to the enrollment date were defined as the “main set” for study 152 purposes" this is confusing and hard to understand, could you please clarify this.

8. Major: line 154-155, how were these outcomes defined and ascertained? Describe it briefly.

9. Major: line 225, "Among non-U.S.–born participants with at least one positive LTBI test result, 34.1% completed LTBI treatment" This is confusing, 2990 out of 3789 completed treatment (78.9%), but the next line refers to 34.1% completing treatment, what's the number? And what do these participants refer to?

10. Major: Table 2, how can the total N of the model presented in the table be 3,973 if in table 1 there is information on homelessness for only 163 participants? Same for other variables e.g., injecting drug use. If what is presented in table 2 is the result of a multivariate analysis and there are missing for those variables, then they will be automatically dropped by the model. I would recommend to add the N for each variable if this is an univariate analysis, for a multivariate analysis the N will equal the number in the variable with the lower number of observations. Same for table 3.

11. Major: Table 2, for variable enrollement reason, the category Non-US born should have not been included in the model because all participants included in the analysis are non-US born, so this is a perfect predictor. Same for table 3.

12. Minor: Table 2 and 3, were variables interpreter, time in US, age, HIV, Diabetes and TST included in the model as categorical variables? If so, what's the reference? For all other categories a reference is provided.

13. Major: I am concerned with the inclusion of categorical variables with more than 3 categories in these models with small sample sizes, these type of variables are normally very problematic, could you provide any information on model fit?

14. Minor: line 302, "their understanding of LTBI might be affected by whether their receive patient" should the last their be replaced by they?

Reviewer #2: Thanks for the opportunity to review this manuscript, which considers the association between interpreter use and LTBI treatment outcomes in the United States. This is an interesting manuscript dealing with an important programmatic question, and I’m pleased to see the authors considering this valuable work.

It is well-recognized that the most substantial losses in the LTBI cascade of care arise early, and investigations into factors affecting treatment uptake are highly important for programs to consider. The role of interpreters, particularly in a setting such as this where there is considerable ethnic and language diversity in the cohort, is crucial and understudied, so this work is well-positioned and useful.

In addition to the primary findings, I also note that other characteristics associated with treatment outcomes are reported. These include factors such as homelessness, which is associated with lower treatment outcomes. These are generally in keeping with existing published data on treatment outcomes and so are in themselves less novel, but helpful for contextualising the cohort here.

Comments

1. The finding that interpreter type was not associated with treatment acceptance is interesting, and probably reassuring with regards to available options in different settings. The authors say that all interpreter types were trained rather than informal; can I clarify whether this is also true for the bilingual clinician interviewers (type 3)? That is, was this group trained/accredited in some way as an interpreter, or should this be taken just to mean that they were both trained as interviewers and also bilingual.

2. Table 1 includes several fields with very small numbers, including a single transgender individual and small numbers of people using intravenous drugs in some fields. It would be common practice to redact such small numbers to avoid potential identification of individuals for ethical reasons, which the authors/editors may consider here.

3. It’s interesting to see that duration of residence in the US was positively associated with LTBI treatment acceptance. This may be intuitive, and associated with other factors allowing prioritisation of LTBI treatment later after migration, but is also associated with decreased risk of reactivation – so there may be some perverse incentives at play worth noting where those at greater risk of future TB are less likely to accept therapy.

4. The steady and linear association between increasing education and decreasing acceptance of LTBI treatment is striking here, and I think worth commenting on further in the text. How would the authors understand this phenomenon?

5. The authors note that the study is limited by lack of objective data on English proficiency. This can’t be changed and is reasonably acknowledged in the text already, but I think a further comment about the human rights need to ensure adequate interpreting services are at least available and accessible for adequate provision of clinical services would be appropriate, perhaps with reference to the existing literature on use of interpreting services in TB programs.

6. PLOS authors have the option to publish the peer review history of their article (what does this mean?). If published, this will include your full peer review and any attached files.

Reviewer #1: No

Reviewer #2: No

---

## [Author Response · Author response to Decision Letter 0]

8 Jan 2024

Reviewers' comments

Reviewer #1: 

The article presents the evaluation of the use of an interpreter and its effect on LTBI treatment offer, acceptance and completion in non-US born individuals. The study uses a subpopulation of non-U.S.–born who participated in a previously published study that assessed LTBI diagnostics and their predictive capabilities to detect progression of those with LTBI to TB disease. This is a very important topic and the manuscript presents results that are of interest to the scientific community, however, the authors must first address a number of issues.

Thank you for your supportive comments and helpful critiques.

1. Major: The title refers to "treatment outcomes" however, I think this title may be misleading because the study does not address LTBI treatment outcomes as they may be understood e.g., disease Vs no disease or cured Vs no cured. As the authors mention in line 296-297 of the discussion section "effect of the use of interpreters on LTBI treatment offer, acceptance, and completion" that's what they evaluated. The title should be modified accordingly. This should also be modified in line 30 of the abstract.

We appreciate this comment. We have changed the title to the following and edited the abstract:

“Interpreter usage and associations with latent tuberculosis infection treatment acceptance and completion in the USA among non-U.S.–born persons, 2012–2017”.

2. Minor: abstract lines 34-38, it is not clear what is a primary outcome and a secondary outcome.

Thank you. We have edited the abstract to include the following sentence, “Our primary outcomes were whether use of an interpreter was associated with LTBI treatment acceptance and completion.”

3. Minor: line 79 of the introduction, please include a reference to support the statement "LTBI treatment is highly effective in preventing progression of LTBI to TB disease"

We have added the following reference in support of this statement: 

Zenner D, Beer N, Harris RJ, Lipman MC, Stagg HR, van der Werf MJ. Treatment of Latent Tuberculosis Infection: An Updated Network Meta-analysis. Ann Intern Med. 2017 Aug 15;167(4):248-255. PMID: 28761946.

4. Major: lines 106-108, Clarify what the primary and secondary outcomes are: Primary outcome seems to be treatment acceptance and completion. Secondary outcomes: association with clinicians decision to offer treatment? Association with treatment outcomes? If that's the primary outcome, as mentioned in point 1, the title of the manuscript should be changed "associations with latent tuberculosis infection treatment acceptance and completion" In lines 181-182 and 184-185 mentions that the primary outcome was acceptance and completion.

We have clarified our primary outcomes in the concluding paragraph of the Introduction with the following, “Given the importance of addressing healthcare inequities and improving outcomes across the LTBI care continuum in non-U.S.–born persons, we investigated associations between the use of trained medical interpreters and LTBI treatment acceptance and completion in a TBESC study as our primary outcomes.”

We have also edited the manuscript title (please see #1, above).

5. Minor: line 109, replace "type associated with" for "type was associated with"

Thank you for catching this error. We have made this edit.

6. Minor: line 125-126, why was a hierarchy of enrollment reasons necessary? There's no further mention to this or enough explanation on why it was important/needed.

Great comment. As “enrollment reason” is a variable included in our regression models, we used this hierarchy to assign participants to a category. We have edited the cited sentence to the following, “For participants with more than one eligibility criterion, a hierarchy of enrollment reasons was established to assign a category to this variable in regression models: 1) close contact of person with infectious TB, 2) non-U.S.–born person from a high-risk country or recently arrived from medium risk country (Supplemental Table 1), 3) visitor of ≥30 days in a high-risk country during prior 5 years, 4) person belonging to a population with a LTBI prevalence ≥25%, and 5) person living with HIV.”

7. Minor: lines 149-152 ”If a participant had more 150 than one TST, QFT-GIT, or T-SPOT test within 14 days, the TB infection test(s) where all three tests were performed closest to the enrollment date were defined as the “main set” for study 152 purposes" this is confusing and hard to understand, could you please clarify this.

We have edited the sentence to the following, “If a participant had more than one result from a type of TB infection test (i.e., TST, QFT-GIT, or T-SPOT) within the 14-day period, the test result performed closest to the enrollment date was used for study purposes.”

8. Major: line 154-155, how were these outcomes defined and ascertained? Describe it briefly.

We have edited the sentence to the following, “The results from decisions to offer LTBI treatment and participant acceptance and completion of treatment were reported to CDC as determined by study site clinic providers and clinic-specific practices.”

9. Major: line 225, "Among non-U.S.–born participants with at least one positive LTBI test result, 34.1% completed LTBI treatment" This is confusing, 2990 out of 3789 completed treatment (78.9%), but the next line refers to 34.1% completing treatment, what's the number? And what do these participants refer to?

We have clarified this sentence, “Among non-U.S.–born participants with at least one positive LTBI test result, 2990 out of 8761 (34.1%) completed LTBI treatment.”

10. Major: Table 2, how can the total N of the model presented in the table be 3,973 if in table 1 there is information on homelessness for only 163 participants? Same for other variables e.g., injecting drug use. If what is presented in table 2 is the result of a multivariate analysis and there are missing for those variables, then they will be automatically dropped by the model. I would recommend to add the N for each variable if this is an univariate analysis, for a multivariate analysis the N will equal the number in the variable with the lower number of observations. Same for table 3.

We appreciate the opportunity to clarify the data. The housing status is known for 8740 participants. We did not include a Table 1 row for the 8598 housed participants. We have corrected this in Table 1 and added similar rows for correctional facility, holding center, and long-term care facility. For clarity we have added the “n” for injection drug use and alcohol consumption, both of which have high levels of missing data. Variables with high levels of missing data (e.g., injection drug) were not included in our models. 

The ”N” values for Tables 2 and 3 are indicated in the Titles, 3973 and 3626, respectively.

11. Major: Table 2, for variable enrollment reason, the category Non-US born should have not been included in the model because all participants included in the analysis are non-US born, so this is a perfect predictor. Same for table 3.

This variable and its categories refer to the primary reason that the participant was enrolled into the TBESC study. This is addressed in our response to question 6, above.

12. Minor: Table 2 and 3, were variables interpreter, time in US, age, HIV, Diabetes and TST included in the model as categorical variables? If so, what's the reference? For all other categories a reference is provided. 

Again, thank you. Time in U.S. and age are continuous variables. The reference for HIV is HIV-uninfected, for diabetes is being without diabetes and for TST positive is having a TST negative result. For these categorical variables, we have added rows to Tables 2 and 3 that clarify the reference categories.

13. Major: I am concerned with the inclusion of categorical variables with more than 3 categories in these models with small sample sizes, these type of variables are normally very problematic, could you provide any information on model fit?

We appreciate this rigorous suggestions. We reviewed the levels for each categorical variable included in our primary outcomes and identified two with small numbers of participants (<10). We found that in the treatment completion model (Table 3), primary enrollment reason of having spent at least 30 days in a high-risk country (n=6) and education level of “other” (n=4) had few participants. Rather than collapsing these categories into other unrelated categories, we have removed these participants, edited table 3 and included footnotes on the table to this effect.

14. Minor: line 302, "their understanding of LTBI might be affected by whether their receive patient" should the last their be replaced by they?

Thank you. We have made the suggested edit.

Reviewer #2 

Thanks for the opportunity to review this manuscript, which considers the association between interpreter use and LTBI treatment outcomes in the United States. This is an interesting manuscript dealing with an important programmatic question, and I’m pleased to see the authors considering this valuable work. It is well-recognized that the most substantial losses in the LTBI cascade of care arise early, and investigations into factors affecting treatment uptake are highly important for programs to consider. The role of interpreters, particularly in a setting such as this where there is considerable ethnic and language diversity in the cohort, is crucial and understudied, so this work is well-positioned and useful.

We appreciate the Reviewer 2’s supportive words.

Comments

1. The finding that interpreter type was not associated with treatment acceptance is interesting, and probably reassuring with regards to available options in different settings. The authors say that all interpreter types were trained rather than informal; can I clarify whether this is also true for the bilingual clinician interviewers (type 3)? That is, was this group trained/accredited in some way as an interpreter, or should this be taken just to mean that they were both trained as interviewers and also bilingual.

Thank you for this question. Specific training as a medical interpreter was not required for bilingual clinician interviewers. We have added the following sentence to the Methods, “Training as a medical interpreter was not required for bilingual study interviewers.”

2. Table 1 includes several fields with very small numbers, including a single transgender individual and small numbers of people using intravenous drugs in some fields. It would be common practice to redact such small numbers to avoid potential identification of individuals for ethical reasons, which the authors/editors may consider here.

We appreciate the concern for identification of study participants. This manuscript was extensively reviewed by CDC who are comfortable with the presentation. This is due, in part, to the fact that the data will not be publicly shared until May, 2024, when a fully deidentified dataset will be made available.

3. It’s interesting to see that duration of residence in the US was positively associated with LTBI treatment acceptance. This may be intuitive and associated with other factors allowing prioritisation of LTBI treatment later after migration, but is also associated with decreased risk of reactivation – so there may be some perverse incentives at play worth noting where those at greater risk of future TB are less likely to accept therapy.

We agree with the Reviewer’s attention to this finding. However, we did not highlight it in the discussion because the association did not reach statistical significance (p-value 0.07) in multivariable analysis.

4. The steady and linear association between increasing education and decreasing acceptance of LTBI treatment is striking here, and I think worth commenting on further in the text. How would the authors understand this phenomenon?

We added the following to the Discussion, “The finding that higher levels of education were associated with lower treatment acceptance was unexpected. We hypothesize that this may be due to misconceptions about test results in the setting of BCG vaccination or about TB occurrence in persons with higher socioeconomic status.”

5. The authors note that the study is limited by lack of objective data on English proficiency. This can’t be changed and is reasonably acknowledged in the text already, but I think a further comment about the human rights need to ensure adequate interpreting services are at least available and accessible for adequate provision of clinical services would be appropriate, perhaps with reference to the existing literature on use of interpreting services in TB programs.

Thank you. We have edited the manuscript’s concluding sentence, “Use of an interpreter, in addition to shorter course regimens for LTBI treatment, increases treatment completion rates among non-U.S.–born persons and is an important intervention for addressing health disparities among persons with limited English proficiency.”

Again, thank you for the careful reviews and please do not hesitate to reach out to me with questions.

---

## [Editor Report · Decision Letter 1]

29 Jan 2024

Interpreter usage and associations with latent tuberculosis infection treatment acceptance and completion in the USA among non-U.S.–born persons, 2012–2017

PONE-D-23-18821R1

Dear Dr. Horne,

We’re pleased to inform you that your manuscript has been judged scientifically suitable for publication and will be formally accepted for publication once it meets all outstanding technical requirements.

Kind regards,

Lisa Kawatsu, PhD

Academic Editor

PLOS ONE
---

## [Editor Report · Acceptance letter]

1 Apr 2024

PONE-D-23-18821R1 

PLOS ONE

Dear Dr. Horne, 

I'm pleased to inform you that your manuscript has been deemed suitable for publication in PLOS ONE. Congratulations! Your manuscript is now being handed over to our production team.

Kind regards, 

on behalf of

Dr. Lisa Kawatsu 

Academic Editor

PLOS ONE